

# Effects of substrate and water depth of a eutrophic pond on the physiological status of a submerged plant, *Vallisneria natans*

Aimin Hao, Sohei Kobayashi, Huilin Huang, Qi Mi and Yasushi Iseri

College of Life and Environmental Sciences, Wenzhou University, Wenzhou, Zhejiang, China

## ABSTRACT

Effects of substrate and water depth on the physiological status of a submerged macrophyte, *Vallisneria natans* (Lour.) H. Hara, were determined by measuring biomarkers in leaves and roots, to understand factors limiting the re-establishment of *V. natans* in urban eutrophic ponds. Ramets of *V. natans* were grown in the laboratory using aquaria containing water and bottom mud from a eutrophic pond and maintained under sufficient light in an incubator. The growth and chlorophyll-a (Chl-a) content of leaves were greater in aquaria with mud than in those with sand, which was used as the reference substrate. The contents of a peroxidation product (malondialdehyde (MDA)) and three antioxidant enzymes (superoxide dismutase (SOD), catalase (CAT), and peroxidase (POD)) in leaves and roots, used as stress biomarkers, changed during the experiment, although differences in these contents between mud and sand were not consistent across the experimental days. To control water depth in the field, ramets of *V. natans* were grown in cages with different substrates (mud and sand) installed at different depths (0.5, 1.2, and 2.0 m) in the pond. The mean light quantum during the experiment decreased with increasing depth, from 79.3 μmol/m$^2$ s at 0.5 m to 7.9 μmol/m$^2$ s at 2.0 m. The Chl-a content in leaves decreased, whereas the MDA content in both leaves and roots increased with increasing water depth. All enzyme activities increased at the beginning and then decreased to the end of the experiment at 2.0 m depth, suggesting deterioration of enzyme activities due to depth-related stress. The MDA content and CAT activity were higher for sand than for mud, whereas the difference in the growth and the leaf Chl-a content between substrates remained unclear in the pond. On comparing the laboratory and field experiments, the leaf Chl-a content was found to be lower and the MDA content and enzyme activities exhibited sharp increase for ramets grown in the pond, even at 0.5 m depth, when compared with those grown in the aquaria. Our results suggest that the bottom mud of the pond is not the major limiting factor in the re-establishment of *V. natans*. Because water depth and light attenuation exerted strong stress on *V. natans*, shallow areas or measures to improve water transparency are required to promote the introduction of *V. natans* in eutrophic ponds for successful restoration in urban areas.

Corresponding author
Sohei Kobayashi, koba@wzu.edu.cn

## INTRODUCTION

Urban ponds often suffer from eutrophication (*Harper, 1992*; *Grimm et al., 2008*; *Smith & Schindler, 2009*). In closed aquatic ecosystems with less renewal of water, excessive nutrient input immediately leads to the overgrowth of phytoplankton. Oxygen depletion is particularly severe at the bottom of such aquatic ecosystems because of respiration by overabundant phytoplankton, decomposition of accumulated organic matter, and limited benthic photosynthesis due to reduced light transparency. Such hypoxic conditions result in the exclusion of animals, such as fish and other aerobic organisms, from the bottom. Submerged plants are often used to improve the ecological status of eutrophic aquatic environments because of their ability to absorb nutrients (*Qiu et al., 2001*; *Hilt et al., 2006*; *Sayer et al., 2010*). *Vallisneria natans* (Lour.) H. Hara is a submerged rooted plant, which is widely distributed in freshwater habitats across the Asia and Australia (*Lowden, 1982*). Owing to its adaptability to a wide range of temperatures and substrates (*Xiong & Li, 2000*; *Xie, An & Wu, 2005*; *Ke & Li, 2006*), ability to absorb significant amounts of nutrients (*Wang et al., 2017*; *Xing et al., 2018*), and allelopathic effects on phytoplankton (*Xian et al., 2006*), *V. natans* has been used to restore many urban ponds (*Yan et al., 1997*; *Qiu et al., 2001*).

Despite their potential to improve degraded aquatic ecosystems, submerged plants, including *V. natans*, have been declining in many freshwater bodies due to habitat degradation (*Hilt et al., 2006*; *Cao et al., 2007*; *Qin et al., 2013*). Restoration works in urban ponds often fail to re-establish stands of submerged plants because of the low survival rate of transplants. High concentrations of nutrients, such as $NH_4^+$, inhibit and impairs the growth of *V. natans* (*Cao et al., 2007*; *Wang et al., 2008*). Light availability, which is regulated by water depth and transparency, is the principal factor restricting the growth of *V. natans* in eutrophic aquatic environments (*Xiong, Hou & Zhong, 2005*; *Xiao, Yu & Wu, 2007*; *Bai et al., 2015*). Substrate quality in terms of texture and chemistry has also been shown to affect *V. natans* (*Xie, An & Wu, 2005*; *Li et al., 2012*; *Bai et al., 2015*). In addition, organic-rich mud under anaerobic conditions, which is typical at the bottom of eutrophic ponds, can inhibit the survival of submerged plants (*Barko & Smart, 1986*; *Wu et al., 2009*; *Silveira & Thomaz, 2015*). Physical disturbances, such as waves and currents in lakes and large water habitats (*Madsen et al., 2001*; *Ellawala, Asaeda & Kawamura, 2013*; *Xu et al., 2016*), can also be a key on the survival of submerged plants. Elucidating the factors limiting the survival of submerged plants is required for effective restoration of urban ponds.

Plants under stress exhibit physiological changes in their cells and organs. Chlorophyll (Chl) is a pigment that absorbs solar energy and accelerates photosynthesis, and its content in leaf tissues is often used as an indicator of photosynthetic ability and plant health. Chl-a absorbs light in the blue and red regions and is the primary photosynthetic pigment, whereas Chl-b absorbs light in slightly different regions and is an accessory pigment supporting Chl-a. Stressful conditions increase the accumulation of reactive oxygen species (ROS), such as superoxide radicals ($O_2^-$), hydroxyl radicals ($OH^-$), and hydrogen peroxide ($H_2O_2$), which can damage cell organelles. Products of lipid peroxidation induced by

ROS, such as malondialdehyde (MDA), are also harmful to cell organelles. ROS accumulation can be controlled by antioxidant enzymes such as superoxide dismutase (SOD), catalase (CAT), and peroxidase (POD). Although the activities of such enzymes increase with ROS, excessive accumulation of ROS and oxidative compounds in cells can reduce their activities, ultimately leading to apoptosis. These peroxidation products and enzyme activities have been used as indicators of stress caused by toxic chemicals or low nutrient availability in *V. natans* (*Wang et al., 2008*, *2012*; *Hao et al., 2011*; *Song et al., 2015*).

To promote ecosystem restoration of urban ponds by introducing *V. natans*, we examined the water quality of an urban pond in Wenzhou City, China, and the effects of water, bottom mud, and pond depth on the physiological status of *V. natans* by in vitro and in situ experiments. We measured the Chl-a and Chl-b contents of leaves as indicators of plant health and of MDA, SOD, CAT, and POD in leaves and roots as indicators of stress.

## MATERIALS AND METHODS

### Study site

The study was conducted at Zhong Shan Park, approximately 1 km south of the Ou River, central Wenzhou City, China (Fig. 1). The study pond was approximately 300 m long in the north-south direction, 20 m wide in the east-west direction, and 2 m deep. Authorization of our field survey and experiment in the park was given by Wang Zhenfeng representing Wenzhou Science and Technology Bureau based on a research project (Water Pollution Control and Treatment Technology Innovation Project under Wenzhou Science and Technology Plan Project: W20170002).

The pond has been restored repeatedly using *V. natans*, which was originally abundant in this area, since 2011. Although the water quality improved after planting *V. natans*, aquatic plants, including *V. natans*, died and disappeared after a few years. Subsequently, an aeration device was installed at the surface of the pond. Despite an improvement in dissolved oxygen (DO), the introduction of *V. natans* failed again in recent years. The bottom mud, which was organic-rich and anaerobic state, has been assumed as a cause of the failure because roots of the dead *V. natans* were dwarf and black-colored.

Our preliminary measurements of the pond water showed that the pH was 7.8–8.3, DO was 6–10 mg/L, water transparency was 0.2–0.3 m, turbidity was 40–80 nephelometric turbidity units (NTU), electric conductivity (EC) was 430–465 µs/cm, total nitrogen (TN) and total phosphorus (TP) were 6.59–8.59 mg/L and 0.41–0.50 mg/L, respectively, and the Chl-a concentration was 30–65 µg/L. The collected bottom mud was black and emitted anaerobic odors, exhibited a total organic carbon (TOC) of 15.4–17.1%, and contained 6.0–7.6 mg of TN and 2.2–3.4 mg of TP per unit dry weight g of mud. The aeration device was located at the southern part of the pond (Fig. 1), where we surveyed the water quality and conducted the field experiment.

### Laboratory experiment

The responses of *V. natans* to the mud and water of the pond were examined in the laboratory. The mud and water were collected from the pond a few days before the

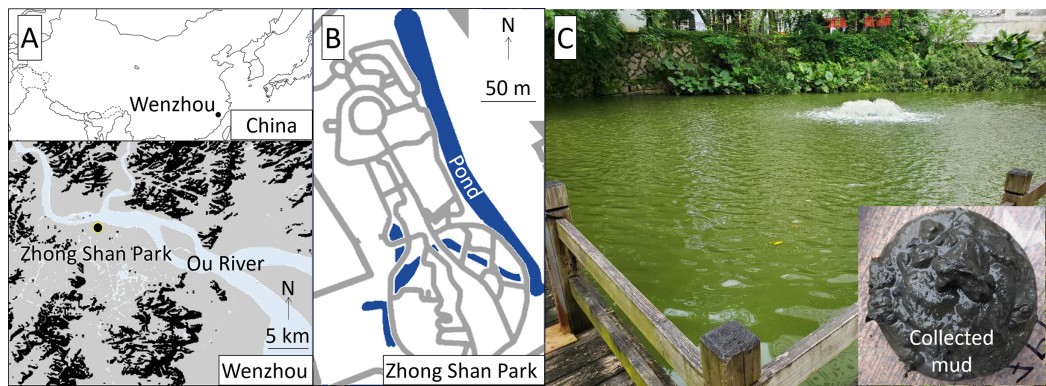

**Figure 1 Location (A), map (B), and view (C) of the study site.**

experiment. We used quartz sand ($SiO_2$: >95%, particle size: 1–2 mm), free of organic matter and nutrients (hereafter we called sand), as the control substrate. Intact and fresh ramets of *V. natans* collected from a rural wetland were used. Five ramets were planted in a 500 mL beaker with a 5 cm layer of the substrate (either mud or sand) at the bottom. Three beakers with ramets were placed in an aquarium ($30 \times 30 \times 50$ cm$^3$, 45 L) containing 30 L of pond water such that the entire propagule was submerged. Three aquaria containing a total of nine beakers and 45 ramets were used for each substrate type.

The aquaria were placed in an incubator maintained at a constant temperature of 25 °C, under a light intensity of 5,000 lux, and light:dark photoperiod of 12 h:12 h for 50 d. For physiological measurements, leaves and roots were sampled from a ramet in each aquarium (i.e., $n = 3$ for each substrate) every 10 d, and then frozen and preserved at −20 °C. A previously unsampled ramet was collected from each aquarium at the end of the experiment for measuring leaf length and number ($n = 3$ for each substrate). Water quality parameters, including temperature, pH, and DO, were measured using a multiparameter water quality meter (Hydrolab DS5X; OTT Hydromet GmbH, Kempten, Germany), and water was sampled for N and P analyses from each aquarium ($n = 3$ for each substrate) at the end of the experiment.

## Field experiment

The responses of *V. natans* to different water depths (0.5, 1.2, and 2.0 m) were surveyed in the southern part of the pond. The mud of the pond and river coarse sand (particle size: 1–2 mm), which was washed to remove organic matter, were used as substrates. The ramets of *V. natans* were planted uniformly in a mesh plastic cage ($40.0 \times 48.5 \times 67.5$ cm$^3$) containing three rectangular trays ($10 \times 25 \times 38$ cm$^3$) with a 7 cm layer of the substrate (either mud or sand) (Fig. 2). A tiered structure was constructed in the pond using steel pipes, and three cages for each substrate type were placed at three different depths (0.5, 1.2, and 2.0 m; Fig. 2).

The experiment commenced on May 11, 2019. The water quality of the pond was evaluated and the ramets of *V. natans* were sampled on 5, 10, 20, and 30 d of the experiment. The vertical profile (0.1 m intervals) of water quality, including temperature,

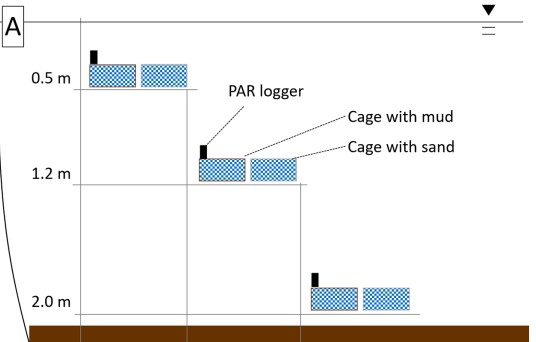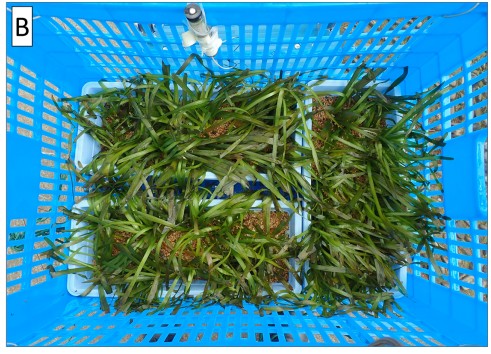

**Figure 2 Schematic view of the field experimental design (A) and arrangement of ramets in a cage (B).**

pH, and DO, was measured near the experimental site using the water quality meter. In addition, water samples (500 mL) were taken from each experimental depth (0.5, 1.2, and 2.0 m) for N and P analyses. The light quantum at each experimental depth was monitored continuously by installing a pocket-size photosynthetically-active radiation logger (DEFI2-L; JFE Advantech Co., Ltd., Nishinomiya, Japan). Water transparency was measured using a Secchi disk. For physiological measurements, leaves and roots were sampled from a ramet in each tray (i.e., $n = 3$ for each depth and substrate combination). A previously unsampled ramet was collected from each tray at the end of the experiment for measuring the leaf length and number ($n = 3$ for each depth and substrate combination).

## Water chemistry and biochemical measurements

Concentrations (mg/L) of N and P were determined according to the Surface Water Environment Quality Standard (GB 3838-2002) (*State Environmental Protection Administration, 2002*). TN and TP were determined by ultraviolet spectrophotometry and the molybdenum blue method, respectively, after digestion of sampled water. Inorganic N ($NH_4^+$-N, $NO_3^-$-N, $NO_2^-$-N) was determined by Nessler's reagent spectrophotometry and ultraviolet spectrophotometry. Chemical oxygen demand (COD) was determined using the potassium dichromate method.

Collected leaves and roots were cut, the surface water was removed using a paper, and the samples were then weighed to obtain the fresh weight. Approximately 2 g of samples were used for evaluating plant health and stress parameters. The Chl content was measured following *Arnon (1949)*. Leaf samples were ground and homogenized with 80% acetone, $CaCO_3$, and quartz sand and then centrifuged at $12,000 \times g$ for 10 min. The supernatant was collected, and its absorbance was measured at 645 and 663 nm. The concentrations of Chl-a and Chl-b were calculated using the following equations:

$$\text{Chl-a} = 12.7A_{663} - 2.69A_{645}$$

$$\text{Chl-b} = 22.9A_{645} - 4.68A_{663}$$

The concentrations were then converted to mg per unit g of leaf fresh weight.

The MDA content was measured following *Heath & Packer (1968)*. Leaf or root samples were ground and homogenized with 5% trichloroacetic acid and quartz sand and then centrifuged at 12,000×g at 4 °C for 10 min. Thiobarbituric acid (2%) was added to the resulting supernatant and the solution was then heated in boiling water for 15 min. After cooling to 20 °C, the solution was again centrifuged at 15,000×g for 10 min. The supernatant was collected and its absorbance was measured at 450, 532, and 600 nm. The concentration of MDA was calculated using the following equation:

$$MDA = 6.45(A_{532}-A_{600})-0.56A_{450}$$

The concentrations were then converted to nmol per unit g of tissue fresh weight.

Prior to the measurement of enzyme activities (SOD, CAT, and POD), the leaf or root samples were ground and homogenized with 50 mM sodium phosphate buffer solution (pH 7.0) and quartz sand and centrifuged at 12,000×g at 4 °C for 20 min. The supernatant was collected immediately for enzyme activity measurements.

The SOD activity was assayed following *Beauchamp & Fridovich (1971)* and *Li, Maezawa & Nakano (2002)*. The reaction solution was prepared by adding 0.1 mL each of 8 mM hydroxylammonium chloride, 3.0 mM EDTA-2Na, 0.15% (w/v) bovine serum albumin (BSA), 8 mM xanthene, and the enzyme extract to 1 mL phosphate buffer (50 mM, pH 7.8). After adding 0.1 mL of xanthine oxidase, the reaction solution was heated for 40 min at 30 °C. One milliliter each of l mL of 20 mM sulfanilic acid and l0 mM N-(1-naphthyl)ethylenediamine dihydrochloride were added to the resulting solution, which was then incubated at 25 °C for 20 min, and the absorbance was measured at 545 nm. One unit of SOD activity was defined as the amount of enzyme required for 50% inhibition of absorbance reduction.

The CAT activity was assayed following *Greenfield & Price (1954)*. The production of $O_2$ from a reaction solution containing 2 mL of 50 mM phosphate buffer (pH 7. 0), 1 mL of the enzyme extract, and 2 mL of 3% $H_2O_2$ was measured by volumetry at normal pressure and 24 °C for 1 min. A unit of CAT activity was calculated by assuming 1 cm$^3$ of $O_2$ as equivalent to 0.041 mmol.

The POD activity was measured following *Kochba, Lavee & Spiegel-Roy (1977)*. A 3 mL of a mixture of 50 mM phosphate buffer (pH 7. 0) and 20 mM guaiacol was added to 0.5 mL of the enzyme extract to prepare the reaction solution. After adding 0.2 mL of 8 mM $H_2O_2$, the absorbance was measured at 470 nm for 1 min.

## Statistical tests

In the laboratory experiment, a split-plot analysis of variance (ANOVA) was performed for leaf Chl-a and Chl-b contents, with substrate (mud, sand) and experiment time (10, 20, 30, 40, 50 d) as fixed factors and aquarium ($n = 3$ for each substrate) as a random factor. The contents of MDA, SOD, CAT, and POD in leaves and roots were analyzed by adding sampled organ (leaf, root) as a fixed factor in the ANOVA. Welch's two-sample $t$-test was used to assess differences in the length and number of leaves between mud and sand collected at the end of the experiment ($n = 3$ for each substrate).

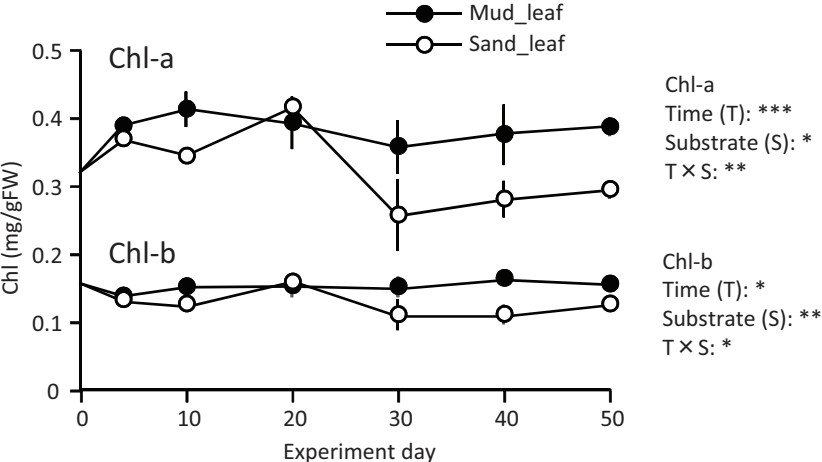

**Figure 3 Changes in the chlorophyll content of leaves in the laboratory experiment.** Error bars denote 1 SD ($n = 3$). Factors with significant effect on Chl-a and Chl-b are shown right (***: $p < 0.001$, **: $p < 0.01$, *: $p < 0.05$). Data of day 0 and 5 were not used in the statistical analysis.

In the field experiment, a split-plot ANOVA was performed for leaf Chl-a and Chl-b contents with depth (0.5, 1.2, 2.0 m), substrate (mud, sand), and experiment time (5, 10, 20, 30 d) as fixed factors and tray ($n = 3$ for each depth and substrate combination) as a random factor. The contents of MDA, SOD, CAT, and POD were analyzed by adding sampled organ (leaf, root) as a fixed factor in the ANOVA. We focused mainly on the effects of substrate and depth, and their interaction with other factors to determine if the effects of substrate or depth varied according to other factors. A two-way ANOVA was done for the length and number of leaves with depth and substrate as fixed factors ($n = 3$ for each depth and substrate combination). Spatio-temporal variation in water quality was also analyzed as a background condition of the experiment. A two-way ANOVA with depth (0.5, 1.2, 2.0 m) and experiment time (5, 10, 20, 30 d) as factors and without replication was performed for variables measured by the water quality meter (temperature, pH, DO, oxidation-reduction potential: ORP, EC, turbidity, and Chl-a concentration), light quantum, nutrients (TN, TP, $NH_4^+$-N, $NO_3^-$-N, and $NO_2^-$-N), and COD of the water samples. For variables measured by the water quality meter, data of the nearest five depths were averaged for each depth (0.5, 1.2, 2.0 m) on each experimental day. For all tests, an α value of 0.05 was used to determine the significance of effects. All statistical analyses were performed in R (version 3.6.3; R Development Core Team, Vienna, Austria), with "lme4" and "lmerTest" packages.

# RESULTS

## Laboratory experiment

A temporal change was detected in the leaf Chl-a content (mg/g), and the effect of the experiment time was significant (Fig. 3; Table 1). The Chl-a content increased slightly from 0 to 10–20 d and then decreased. The decrease from 10 to 20 d to the end of the experiment was greater in aquaria with sand than mud, and the effects of substrate and

**Table 1 Results of split-plot-design analysis of variance (ANOVA) showing the effect of each factor on the physiological indices in the laboratory experiment.**

| Factor | df | Chl-a | | Chl-b | | MDA | | SOD | | CAT | | POD | |
|---|---|---|---|---|---|---|---|---|---|---|---|---|---|
| | | F | p | F | p | F | p | F | p | F | p | F | p |
| Time (T) | 4 | 11.6 | *** | 3.44 | * | 12.7 | *** | 24.1 | *** | 4.54 | ** | 22.5 | *** |
| Organ (O) | 1 | | | | | 463 | *** | 19.2 | *** | 132 | *** | 88.4 | *** |
| Substrate (S) | 1 | 20.2 | * | 39.3 | ** | 0.88 | | 0.27 | | 0.02 | | 1.81 | |
| T*O | 4 | | | | | 3.76 | * | 13.2 | *** | 1.45 | | 9.90 | *** |
| T*S | 4 | 5.03 | ** | 4.73 | * | 2.95 | * | 1.63 | | 2.11 | | 2.81 | * |
| O*S | 1 | | | | | 25.2 | *** | 0.19 | | 1.84 | | 11.1 | ** |
| T*O*S | 4 | | | | | 2.35 | | 0.56 | | 1.45 | | 3.91 | ** |

Note:
Significance of effects are shown by asterisks (***: $p < 0.001$, **: $p < 0.01$, *: $p < 0.05$). Organ and interaction of organ and other factors were not included in the ANOVA for Chl-a and Chl-b.

time × substrate interaction were significant. Consequently, the Chl-a content at 50 d was 19% increase from its 0 d for the mud, and it was 9% decrease from its 0 d for the sand.

Temporal changes in the Chl-b content were relatively small when compared to those in the Chl-a content. However, some patterns were similar to those observed for the Chl-a content, such as significantly higher for mud than for sand, and a significant effect of time × substrate interaction (Fig. 3; Table 1).

The MDA content (nmol/g) of both leaves and roots increased at the beginning of the experiment and then decreased, and it was significantly higher in leaves than in roots (Fig. 4; Table 1). The MDA content decreased visibly after 30 d for leaves, while it decreased steadily after 10 d for roots, and the effect of time × organ interaction was significant. The MDA content of leaves was higher in aquaria with mud than sand, whereas an opposite pattern was observed for roots, and the effect of organ×substrate interaction was significant. The decline in MDA content from 30 to 50 d was steeper for mud than for sand. Consequently, the MDA content at 50 d was 17% (leaves) or 5% (roots) decrease from its 0 d for the mud, and it was 8% (leaves) or 52% (roots) increase from its 0 d for the sand.

The SOD activity (unit/g) increased after 10 d in leaves, whereas it changed less throughout the experiment in roots, and the effect of time×organ interaction was significant (Fig. 4; Table 1). As a consequence, the SOD activity in leaves exceeded that in roots from 20 d onwards. The SOD activity in leaves changed less after 30 d. No significant difference was detected in the SOD activity between the two substrates.

The CAT activity ($H_2O_2$ nmol/g/min) of both leaves and roots increased at the beginning of the experiment and then gradually decreased, and the effect of time was significant (Fig. 4; Table 1). The CAT activity was significantly higher in leaves than in roots throughout the experiment. No significant difference was detected in the CAT activity between the two substrates. The CAT activity at 50 d was 27–54% increase from its 0 d in leaves, and it was 5–10% decrease from its 0 d in roots.

The POD activity ($A_{470}$/g/min) of both leaves and roots increased at the beginning of the experiment and then gradually decreased, and the effect of time was significant

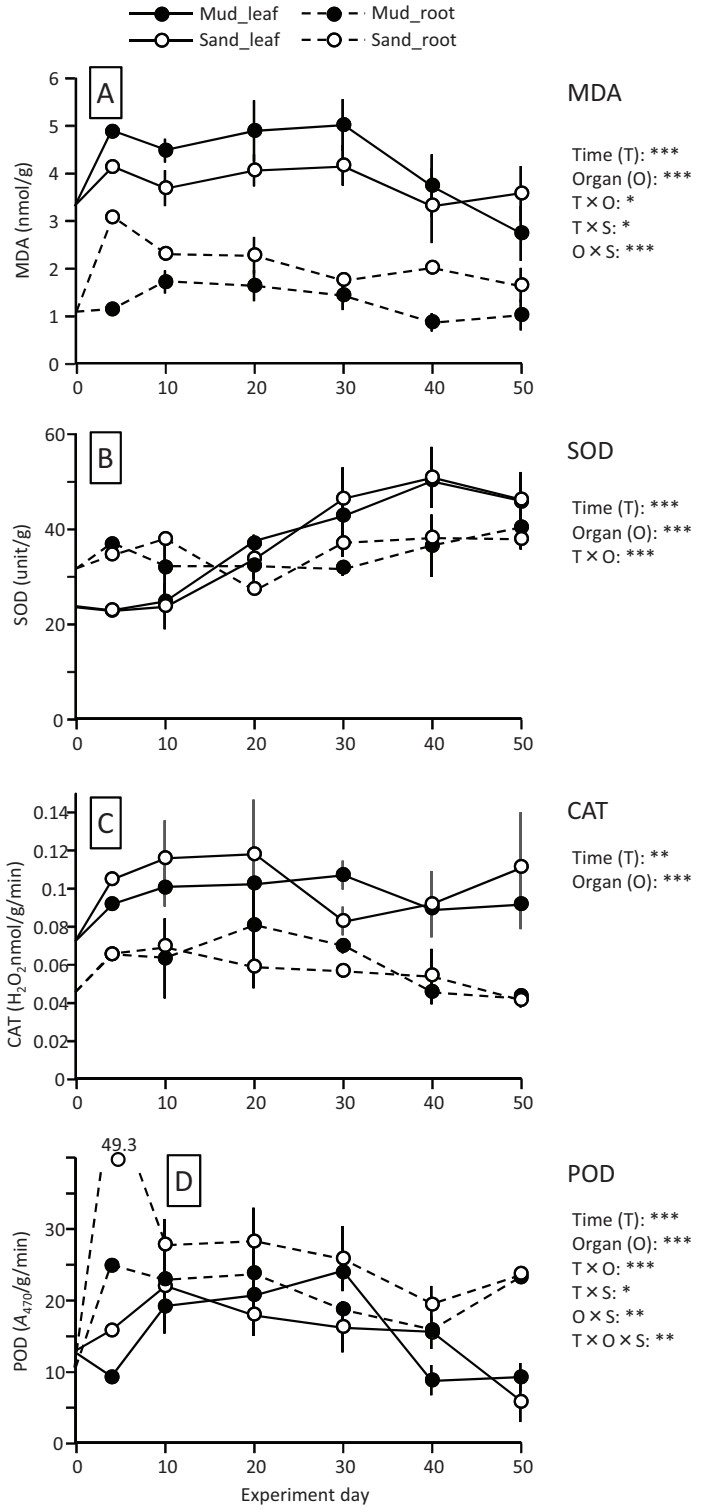

**Figure 4 Changes in lipid peroxidation product (MDA) (A) and enzyme (SOD, CAT, and POD) activity (B–D) of leaves and roots in the laboratory experiment.** Error bars denote 1 SD ($n = 3$). Factors with significant effect on MDA, SOD, CAT, and POD are shown right (***: $p < 0.001$, **: $p < 0.01$, *: $p < 0.05$). Data of day 0 and 5 were not used in the statistical analysis.

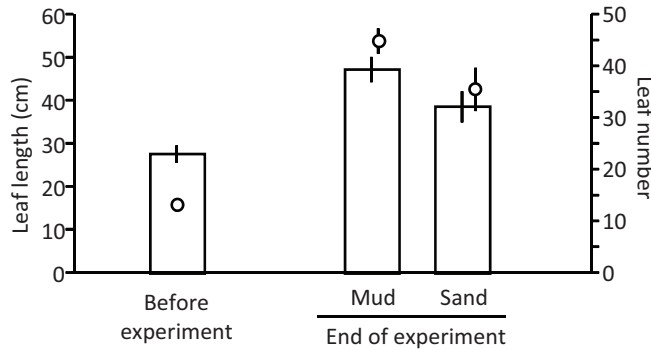

**Figure 5 Length (bar) and number (circle) of leaves before and at the end of the laboratory experiment.** Error bars denote 1 SD ($n$ = 3).

(Fig. 4; Table 1). The POD activity was significantly higher in roots than in leaves throughout the experiment. The POD activity in roots was higher for sand than for mud throughout the experiment, whereas the difference between the two substrates remained unclear in leaves, and the effect of organ×substrate was significant. The POD activity at 50 d was 28–55% decrease from its 0 d in leaves, while it was 120–125% increase from its 0 d in roots.

Ramets grew well in aquaria with water and mud from the pond (Fig. 5). The leaf length (cm) increased from 27.8 cm (±2.0 SD, $n$ = 3) before the experiment to 47.4 cm (±2.9 SD) and 38.8 cm (±3.6 SD) in aquaria with mud and sand, respectively, at the end of the experiment (Welch's two-sample $t$-test, $t$ = −2.720, df = 3.224, $p$ = 0.067). The leaf number per ramet increased from 13.3 (±1.2 SD, $n$ = 3) before the experiment to 45.0 (±2.4 SD, $n$ = 3) and 35.7 (±4.2 SD, $n$ = 3) in aquaria with mud and sand, respectively, at the end of the experiment (Welch's two-sample $t$-test, $t$ = −2.521, df = 3.124, $p$ = 0.083). Thus, the growth of *V. natans* was greater for mud than for sand.

The water quality of aquaria also changed during the experiment. For example, pH increased from 8.3 before the experiment to 9.2 and 9.4, and DO (mg/L) increased from 2.04 to 9.11 and 10.02 in aquaria with mud and sand, respectively, at the end of experiment. Such an increase was expected as a result of plant photosynthesis, which involves consumption of $CO_2$ and production of $O_2$. On the other hand, the concentrations of N and P had decreased after the experiment. For example, TN (mg/L) decreased from 8.59 to 2.34 (±0.65 SD) and 1.83 (±0.52 SD), and TP (mg/L) decreased from 0.11 to 0.078 (±0.017 SD) and 0.071 (±0.010 SD) for mud and sand, respectively. The reduction of N and P in the water seems to be associated with nutrient uptake by *V. natans*. TN and TP were slightly higher for mud, which originally contained nutrients, than for sand.

## Field experiment

The water quality of the pond varied among the sampling days (Table S1). For example, at 0.5 m depth, the water temperature, pH, DO, Chl-a concentration, and turbidity varied from 21.7 to 25.0 °C, 7.9–8.3, 6.4–10.0 mg/L, 28.1–62.9 mg/L, and 34.8–75.8 NTU, respectively. In contrast, the water quality exhibited less variation among 0.5 m, 1.2 m, and

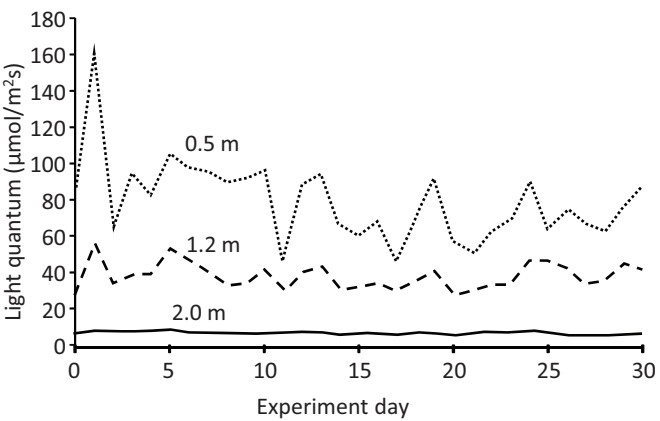

**Figure 6 Light quantum (μmol/m²s) at different depths in the pond during the field experiment.**

2.0 m depths at each day. For example, at 0 d, the water temperature and pH at all depths were 21.7 °C and 8.2, respectively, DO, Chl-a concentration, and turbidity varied from 9.9 to 10.0 mg/L, 62.9–70.4 mg/L, and 44.9–48.6 NTU, respectively. Similarly, the COD (range: 24–77 mg/L), TN (5.63–9.65 mg/L), TP (0.36–0.64 mg/L) varied among days, but they exhibited less variation among the three depths at each day.

The light quantum (μmol/m² s) at each depth also exhibited temporal variation (Fig. 6). In addition, it decreased with increasing depth every day, declining to almost half and less than one-tenth from the depth of 0.5 m (mean: 79.3 μmol/m² s) to 1.2 m (38.3 μmol/m² s) and 2.0 m (6.7 μmol/m² s), respectively. Water transparency gradually increased from 0.25 to 0.30 m during the experiment (Table S1). Rainy days were more frequent in the latter half of the experiment, with a maximum daily rainfall of 42.2 mm and total rainfall of 208 mm during the experiment (Fig. S1).

The leaf Chl-a content (mg/g) declined sharply at the beginning of the experiment, particularly at the depths of 1.2 and 2.0 m (Fig. 7). The change in Chl-a content after 10 d differed depending on the water depth, and the effects of time and time×depth interaction were significant (Fig. 7; Table 2). After 10 d, the Chl-a content increased slightly at the depth of 0.5 m, whereas it declined steadily at 1.2 and 2.0 m. The Chl-a content decreased with increasing depth, and the effect of depth was significant. However, no clear difference was observed between mud and sand. The Chl-b content also decreased with increasing depth, and the difference between mud and sand remained unclear (Fig. 7; Table 2).

The MDA content (nmol/g) of leaves increased 2- to 3-fold during the experiment, whereas the content of roots changed less, and the effects of time and time×organ interaction were significant. The MDA content was significantly higher in leaves than in roots throughout the experiment (Fig. 8; Table 2). The MDA content was also significantly higher in deeper positions, and significantly higher for sand than for mud. Consequently, the MDA content of leaves and roots at 30 d was the lowest for mud at 0.5 m.

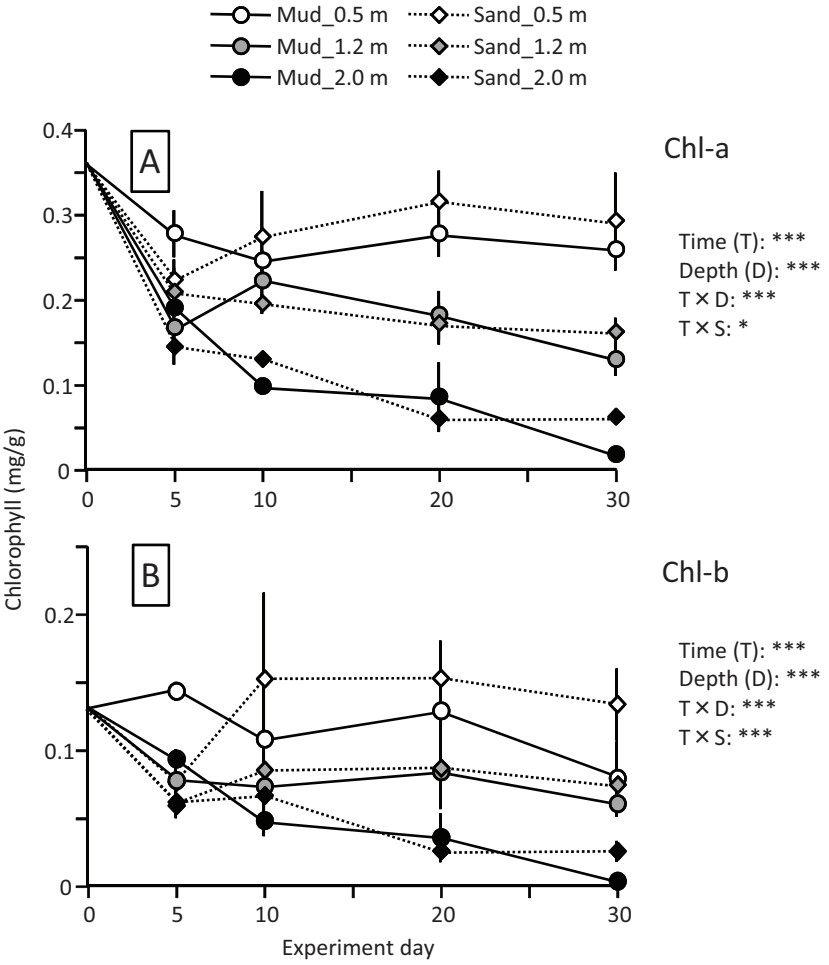

**Figure 7 Changes in Chl-a (A) and Chl-b (B) contents of leaves during the field experiment.** Error bars denote 1 SD (*n* = 3). Factors with significant effect on Chl-a and Chl-b are shown (***: $p < 0.001$, **: $p < 0.01$, *: $p < 0.05$). Data of day 0 were not used in the statistical analysis.

The SOD activity (unit/g) of leaves and roots increased at the beginning of the experiment at all depths. However, the change of the SOD activity in the latter half of the experiment differed depending on the depth (Fig. 8; Table 2); during this period the SOD activity at 0.5 and 1.2 m increased continuously but slowly, whereas that at 2.0 m decreased, and the effects of time and time×depth interaction were significant. At 0.5 and 1.2 m, the SOD activity increased 3- to 4-fold and 2- to 3-fold during the experiment in leaves and in roots, respectively, and the effects of organ, time × organ interaction, and organ × depth interaction were significant. In addition, the SOD activity was significantly higher for mud than for sand, particularly in roots at 0.5 and 1.2 m depths, and the effects of substrate and time×substrate, organ × substrate, and depth×substrate interactions were significant.

The CAT activity ($H_2O_2$ nmol/g/min) of leaves and roots increased at the beginning of the experiment at all depths. However, the change of the CAT activity in the latter half of the experiment differed depending on the depth (Fig. 8; Table 2); during this period, the

**Table 2 Results of the split-plot-design analysis of variance (ANOVA) that show the effect of each factor on the physiological indices in the field experiment.**

| Factor | df | Chl-a | | Chl-b | | MDA | | SOD | | CAT | | POD | |
|---|---|---|---|---|---|---|---|---|---|---|---|---|---|
| | | F | p | F | p | F | p | F | p | F | p | F | p |
| Time (T) | 3 | 12.0 | *** | 6.48 | *** | 92.0 | *** | 121 | *** | 118 | *** | 37.5 | *** |
| Organ (O) | 1 | | | | | 422 | *** | 85.3 | *** | 0.53 | | 213 | *** |
| Depth (D) | 2 | 266 | *** | 89.4 | *** | 46.3 | *** | 43.4 | *** | 87.8 | *** | 44.0 | *** |
| Substrate (S) | 1 | 1.42 | | 1.30 | | 9.70 | ** | 23.1 | *** | 223 | *** | 1.32 | |
| T*O | 3 | | | | | 36.3 | *** | 12.2 | *** | 4.99 | ** | 12.3 | *** |
| T*D | 3 | 12.4 | *** | 5.21 | *** | 2.02 | + | 103 | *** | 142 | *** | 42.9 | *** |
| T*S | 6 | 3.70 | * | 11.0 | *** | 6.60 | *** | 8.03 | *** | 3.16 | * | 2.63 | |
| O*D | 2 | | | | | 0.52 | | 20.9 | *** | 0.66 | | 4.22 | * |
| O*S | 1 | | | | | 20.9 | *** | 6.21 | * | 5.07 | * | 1.11 | |
| D*S | 2 | 0.25 | | 0.84 | | 0.14 | | 6.09 | * | 7.93 | ** | 1.92 | |

**Note:**
Significance of effects are shown by asterisks (***: $p < 0.001$, **: $p < 0.01$, *: $p < 0.05$, +: $p < 0.1$, results of the interaction among three or more variables were omitted). Organ and interaction of organ and other factors were not included in the ANOVA for Chl-a and Chl-b.

CAT activity at 0.5 and 1.2 m increased continuously but slowly, whereas that at 2.0 m decreased. This pattern was similar to that of the SOD activity. The difference in the CAT activity between leaves and roots remained unclear. The CAT activity was significantly higher for sand than for mud.

The POD activity ($A_{470}$/g/min) of leaves and roots increased slightly at 0.5 and 1.2 m during the experiment, whereas it increased sharply at the beginning of the experiment and then decreased at 2.0 m depth, and the effects of time and time×depth interaction were significant (Fig. 8; Table 2). The POD activity was significantly higher in leaves than in roots, whereas no significant difference was observed between mud and sand.

The ramets of *V. natans* did not exhibit an apparent increase in the length and number of leaves during the experiment (Fig. 9); they rather decreased at 1.2 m and 2.0 m depths. The length and number of leaves tended to decrease with increasing water depth, and were less for sand than for mud, but due to a variation among trays a significant effect was detected only for depth and depth × substrate interaction on leaf number (depth: df = 2, $F = 24.4$, $p < 0.001$; substrate: df = 1, $F = 3.56$, $p = 0.071$; depth × substrate: df = 2, $F = 5.06$, $p = 0.015$). The leaves of ramets at 0.5 m were completely green, but some leaf apices were flat (i.e., not acute like before the experiment), resembling being cut by animals in the pond (e.g., fish, birds). The leaves of ramets at 1.2 m were only partially green, and the leaves at 2.0 m were light brown, indicating senescence.

## DISCUSSION

In the present study, we examined the growth and physiological status of *V. natans* under the water and mud conditions of an urban pond using laboratory and field experiments. Plant physiological status was assessed using photosynthetic pigments, Chl-a and Chl-b, as plant health indicators, and a lipid peroxidation product, that is, MDA, and

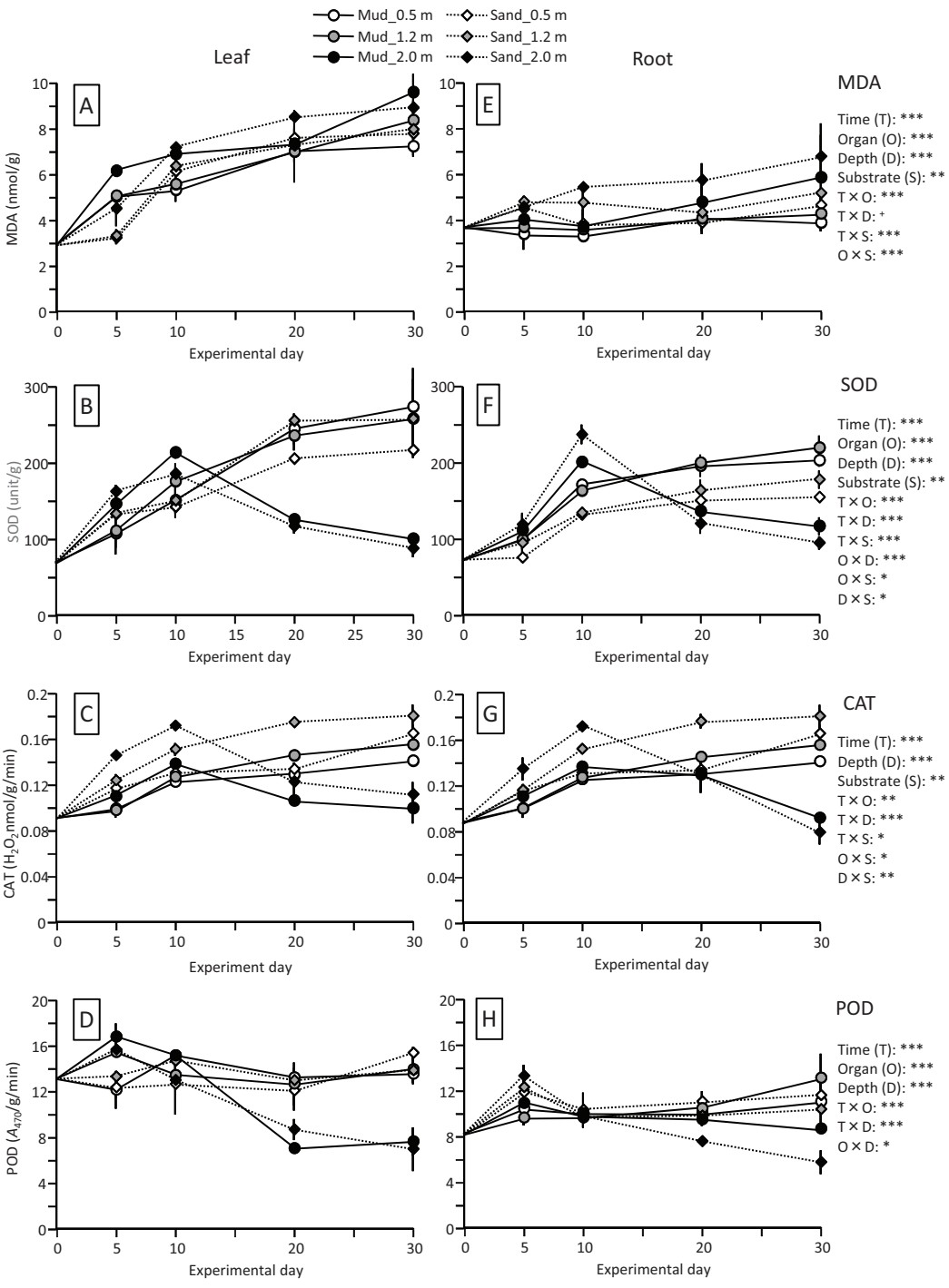

**Figure 8 Changes in lipid peroxidation product (MDA) and enzyme (SOD, CAT, and POD) activity of leaves (A–D) and roots (E–H) in the field experiment.** Error bars denote 1 SD ($n = 3$). Factors with significant effect on MDA, SOD, CAT, and POD are shown (***: $p < 0.001$, **: $p < 0.01$, *: $p < 0.05$, +: $p < 0.1$). Data of day 0 were not used in the statistical analysis.

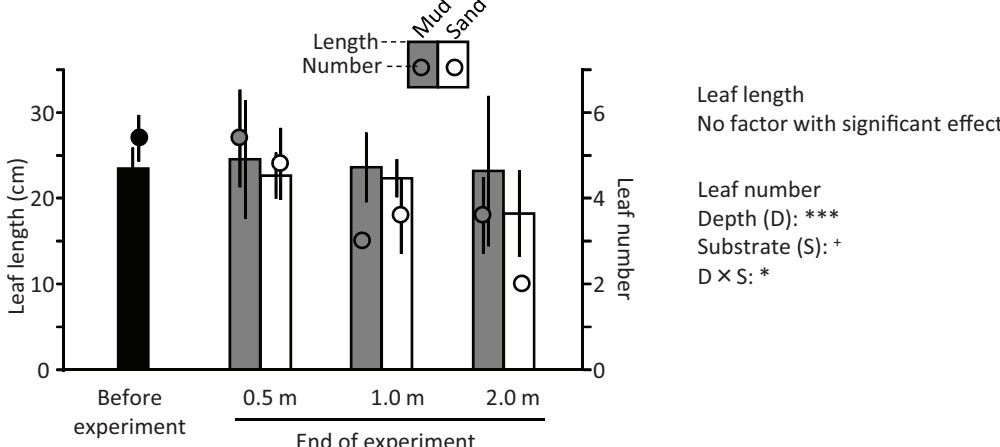

**Figure 9 Length (bar) and number (circle) of leaves before and at the end of the field experiment.** Error bars denote 1 SD ($n = 3$). Factors with significant effect on length and numbers of leaves are shown (\*\*\*: $p < 0.001$, \*\*: $p < 0.01$, \*: $p < 0.05$, +: $p < 0.1$). Data of before experiment were not used in the statistical analysis.

antioxidant enzymes, including SOD, CAT, and POD, as stress indicators. Ramets grew well in the laboratory and physiological status differed less between 0 and 50 d of the experiment in both mud and sand. Thus, the water of the pond, which is more or less polluted due to urban human activities, is unlikely to directly or adversely affect *V. natans*. High N and P concentrations can have toxic effects on *V. natans* (*Cao et al., 2007*; *Wang et al., 2008*). However, N and P concentrations in water decreased during our laboratory experiment, which is likely to be associated with nutrient uptake and growth of *V. natans*. We also revealed that the growth and physiological status of *V. natans* were better with mud than with sand as a substrate. Thus, although the mud of the pond was rich in organics, an adverse effect of such mud on submerged plants (*Barko & Smart, 1986*; *Wu et al., 2009*; *Silveira & Thomaz, 2015*) was not evident in this study. We used sand, which was free of organic matter, as the control substrate. Substrate type and nutrient contents are considered to be important for the nutrient acquisition by the roots of *V. natans* (*Xie, An & Wu, 2005*; *Xiao, Yu & Wu, 2007*; *Bai et al., 2015*). Although high N and P concentrations in water possibly compensate for a low nutrient content in substrate, sand may be less suitable than natural mud or clay for roots to adjust and acquire nutrients. Further studies using different substrates, such as mud in oligotrophic lakes, are required to examine the suitability of the mud from eutrophic ponds for transplanting *V. natans*.

The adverse effects of water depth and reduced light availability on *V. natans* were evident in the field experiment. The leaf number of ramets decreased with increasing depth at the end of the experiment. Moreover, the Chl-a and Chl-b contents of leaves decreased, and the MDA content of leaves and roots increased with increasing water depth. Interestingly, the activities of antioxidant enzymes, SOD, CAT, and POD, increased at the beginning of the experiment and then decreased at 2.0 m depth. This was likely due to the oxidative damage caused by excessive production of ROS under low-light conditions at
this depth. Such a deterioration of enzyme activity has been reported for *V. natans* exposed to lead (*Yan et al., 2006*; *Wang et al., 2012*). Light availability is the important factor that varies vertically in the study pond. A sharp decline in light availability with increasing depth was detected in this study; the light quantum was reduced by 50% and 90% from 0.5 m to 1.2 and 2.0 m, respectively. *Bai et al. (2015)* also reported a 62% and 99% reduction in the light quantum from the surface to 0.6 and 1.8 m in an experimental pond, respectively. Such a sharp attenuation in light availability is typical of eutrophic ponds and lakes (*Hodoki & Watanabe, 1998*; *De Lange, 2000*). On the other hand, only small differences were detected in the vertical profile of water quality, including temperature, DO, and N and P concentrations, in this pond. These results suggest that *V. natans* experiences strong stress at greater depths in this pond due to light depletion, which impedes photosynthesis. Most area inside the study pond was near or greater than 2.0 m deep, at which the growth of *V. natans* is likely to be inhibited.

Light availability, which is regulated by water depth and transparency, has been identified as the main factor limiting the distribution of submerged plants including *V. natans* (*Voesenek et al., 2006*; *Bai et al., 2015*; *Dong et al., 2004*; *Han & Cui, 2016*). The optimal water depth for the growth of *V. natans* has been reported as 100–160 cm in an oligotrophic lake (*Xiao, Yu & Wu, 2007*), in which water transparency and light availability were high. However, a steep light attenuation with increasing depth in eutrophic lakes and ponds is likely to limit the distribution of *V. natans* to shallow areas. *Han & Cui (2016)* used the ratio of transparency to water depth as an indicator of eutrophication pressure on macrophyte communities. They suggested that the ratio should be no less than 0.52 to restore submerged species in eutrophic ponds. Based on this criterion, and because the maximum transparency in our study pond was 0.3 m, shallow area less than 0.58 m deep are required to restore submerged plants. Although a deterioration of enzyme activity was not observed at 0.5 and 1.2 m in this study, *V. natans* is unlikely to be able to tolerate low light stress for a long period. Less growth, reduced Chl contents, and increased level of stress indicators of *V. natans* even at 0.5 m in this study may suggest that 0.5 m or shallower areas are required for the growth of this plant in this pond. However, too shallow area may be unsuitable for *V. natans*, which is originally adapted to habitats deeper than 1 m (*Xiao, Yu & Wu, 2007*; *Li et al., 2020*), to grow vertically well in the water column. Because we examined limited depths deeper than 0.5 m in the field, further studies are needed to clarify the suitability of such shallow areas for the growth and survival of *V. natans* in eutrophic ponds.

The strong effect of depth on *V. natans* might have obscured the differences between mud and sand in the field experiment. Differences in growth and leaf Chl-a and Chl-b contents between mud and sand, which were observed in the laboratory experiment, were not detected in the field experiment. The MDA content and CAT activity of leaves and roots were higher for sand than for mud in the field, which were not detected in the laboratory experiment. Although there are different results between the experiments, the results of field study suggest that the mud of the pond is unlikely to be a limiting factor in the establishment of *V. natans*.

Sufficient light and low nutrient concentrations in water were more advantageous to the ramets in the laboratory than to those in the field. The growth and physiological status of leaves and roots were obviously better in the laboratory than in the field, evident from the increase in length and number of leaves, lesser decrease in the Chl-a and Chl-b contents, and lesser increase in the MDA content and antioxidant enzyme activities. The MDA content and activities of antioxidant enzymes, except SOD, declined after a small increase in the early stages of the laboratory experiment. The decline of these indices to the initial or even smaller values at the end of the experiment may indicate ramets acclimatization to the aquaria environment. The mean light quantum at 0.5 m depth (79.3 μmol/m$^2$ s) was similar to the intensity of light in the incubator (70–80 μmol/m$^2$ s). However, the light quantum would be smaller inside the meshed cages used to grow ramets in the field. Although the nutrient concentration was initially similar between the aquaria and pond, nutrient uptake by ramets substantially reduced the nutrient concentration in the aquaria at the end of the experiment. High nutrient concentrations can adversely affect *V. natans*, both directly and indirectly, by promoting epiphytic algal growth on *V. natans* (*Song et al., 2015*). Ramets in the field were also at a risk of grazing by animals such as birds and fish. Some leaf apices of *V. natans* were flat, likely to have been eaten partially by pond animals. This was supported by a chance observation of a fish feeding on *V. natans* collected in a bucket (H. Huang, 2019, personal observation). The better growth and physiological status of ramets in the laboratory than in the field may be partially associated with the absence of predator and hydrologic disturbance in the former. Present study is inadequate to show the importance of water depth and light on *V. natans* across different seasons and life stages, and how physical disturbances can modify the depth related responses of *V. natans*.

Previous studies have shown increased MDA content in *V. natans* growing under salinity, lead, and ammonia stress (*Wang et al., 2008*; *Hao et al., 2011*; *Li et al., 2011*; *Song et al., 2015*). In our study, the MDA content was higher with increasing depth. The accumulation of MDA eventually inactivates the enzymes associated with photosynthesis, respiration, and other metabolic processes in plant cells (*Song et al., 2015*). Of the three enzymes analyzed in this study, SOD showed the strongest response, in terms of the magnitude of changes in both laboratory and field experiments. SOD converts $O_2^-$ into $O_2$ and $H_2O_2$ in the first step of ROS removal (*Apel & Hirt, 2004*; *Rahnama & Ebrahimzadeh, 2005*). On the other hand, POD exhibited the fastest response (i.e., stopped increasing the earliest) among the enzymes in both laboratory and field experiments. Both POD and CAT convert $H_2O_2$ into $H_2O$ and $O_2$ (*Bowler, Montagu & Inzé, 1992*). Responses of these enzymes varied depending on the study (*Yan et al., 2006*; *Hao et al., 2011*; *Li et al., 2011*; *Wang et al., 2012*), and further studies are required to generalize the response of each enzyme.

## CONCLUSIONS

We demonstrated important factors on the growth and physiological status of *V. natans* in eutrophic urban ponds using laboratory and field experiments. Sufficient light availability is required for better physiological status of the species. Owing to the sharp attenuation

of light with increasing depth, shallow areas less than 1 m deep and improved water transparency are fundamental requirements for successful re-introduction of *V. natans* in eutrophic ponds. Despite the anaerobic conditions prevailing in the mud at the pond bottom, no adverse effects were detected on *V. natans* in the present study. Thus, it is suggested that the current status of the bottom mud does not directly inhibit the growth of submerged species. However, it can indirectly affect growth by releasing nutrients in the water column, which, in turn, can induce algal blooms. Efforts to reduce the nutrient load are also important to limit the phytoplankton overgrowth, and thus, to maintain transparency and ensure light availability to submerged plants.

## ACKNOWLEDGEMENTS

We thank the employes of the Wenzhou Science and Technology Bureau and Wenzhou Park Management Office for providing valuable information and support throughout the field study. We also thank the students of Aquatic Ecology and Symbiology, College of Life and Environmental Sciences, Wenzhou University, China, for helping with the laboratory and field surveys. The manuscript has been greatly improved by two reviewers. Finally, we would like to thank Editage for English language editing.

### Funding

This research was funded by the Wenzhou Science and Technology Bureau according to the Water Pollution Control and Treatment Technology Innovation Project under Wenzhou Science and Technology Plan Project (W20170002, Research and development of new technology to improve the water environment by using the natural ecological water purification function). The funders had no role in study design, data collection and analysis, decision to publish, or preparation of the manuscript.

### Grant Disclosures

The following grant information was disclosed by the authors:
Wenzhou Science and Technology Plan Project: W20170002.

### Competing Interests

The authors declare that they have no competing interests.

### Author Contributions

- Aimin Hao conceived and designed the experiments, performed the experiments, prepared figures and/or tables, authored or reviewed drafts of the paper, and approved the final draft.
- Sohei Kobayashi analyzed the data, prepared figures and/or tables, authored or reviewed drafts of the paper, and approved the final draft.
- Huilin Huang performed the experiments, analyzed the data, prepared figures and/or tables, authored or reviewed drafts of the paper, and approved the final draft.

![PeerJ]

- Qi Mi performed the experiments, analyzed the data, prepared figures and/or tables, and approved the final draft.
- Yasushi Iseri conceived and designed the experiments, performed the experiments, prepared figures and/or tables, and approved the final draft.

## Field Study Permissions

The following information was supplied relating to field study approvals (i.e., approving body and any reference numbers):

Field experiment in Zhong Shan Park was approved by the Wenzhou Science and Technology Bureau according to the research project (Water Pollution Control and Treatment Technology Innovation Project under Wenzhou Science and Technology Plan Project: W20170002).

## Data Availability

The raw data are available in the Supplemental Files.

## Supplemental Information

Supplemental information for this article can be found online at http://dx.doi.org/10.7717/peerj.10273#supplemental-information.

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
