# Peer review of "Effects of substrate and water depth of a eutrophic pond on the physiological status of a submerged plant, Vallisneria natans"

_PeerJ, doi:10.7717/peerj.10273_

## Round 0.1 · original submission · Major Revisions

Two reviewers as experts in your field have highlighted the strengths of your submitted manuscript and have proposed a number of changes to improve the quality. Please consider the proposed changes of both reviewers as mandatory for a reconsideration of your revised manuscript for publication in PeerJ.

I look forward to your revised manuscript.

·

Basic reporting

Please see the attachment.

Experimental design

Please see the attachment.

Validity of the findings

Please see the attachment.

Additional comments

In the manuscript entitled: Effects of substrate and water depth of a eutrophic pond on the physiological status of Vallisneria natans, a useful aquatic plant for water environment restoration in urban areas” authors explore the limiting factor for the growth of V. natans in eutrophic urban ponds. They measure various compounds in root and shoot to assess the stress level of V. natans when grown under different depth/light conditions and substrates.
General comments
1- The title can be more concise. Try removing the second part: a useful aquatic plant…
2- The study is conducted using a small laboratory experiment and then in the field using a eutrophic urban pond. But results and discussion on these two parts of the study are not well organized. The text needs to be reworked and reorganized to clarify what authors have found from each experiment separately and then what the pilot study (lab exp) contributed to the field study.
3- Tables and figures are not consistent. They are not clear and lack the information necessary for the assessment of research findings. Most importantly, statistical analysis needs to be incorporated into the figures or shown to clearly demonstrate the differences across depths and substrates.
4- Statistical analysis is not explained properly. With two experiments (lab and field) and different experimental designs, authors need to provide a complete explanation of how and why they decided to run the stats with the current method.
5- Several compounds are measured and discussed in the text based on leaf analysis and the majority of the results and discussion are focused on these compounds without a word of how the plants actually performed at different treatments/depths. only plant assessment for lab exp was shown which seems unrealistic. The data are perhaps rounded up and somehow all stnd dev are 5! In addition, there is no description of how the authors assessed these data (height, fresh weight, etc.)
6- Substrate is a big component of this study and it is even in the title. But there is no information on substrate texture and chemistry.

This manuscript can be significantly improved by English editing and requires careful attention to the disclosed comments.

Abstract
20 it is better to define “physiological status” or be more specific. Please state what physiological traits you have measured.
22 “water environmental conditions” sounds odd. Throughout the text.
26 “different physiological status” is vague. You could change the sentence to” MDA, SOD, CAT and POD content differed in leaves and roots when plants were grown in different substrates…”.
35 due to environmental stress. This is too general. If your treatments are depths, just make statements about the depth to avoid generalization.
34-38 needs reorganizing to make it clear what is compared to aquaria or sand. Try to separate the findings between the laboratory and field experiment to avoid confusion.
40 bottom mud has adverse effect?! Why is that expected? Please explain.
42 this study talks about water depth and ties that with light attenuation. But there is no mention to light intensity at different depths in the abstract.
53 please avoid using the word “purify”. Plants do not purify the water.
63 “because of less survival of planted individuals” sounds odd. Maybe change to ‘ because of low survival rate of transplants or introduced plants”
64 inhibits
69 shorten the sentences, here: . but it is not clear whether…
70-72 I do not understand the connection between hydraulic forces of waves in large water bodies and lakes to this study on small urban ponds. I suggest removing this statement.
75-76 you measured both Chl a and b. explain their difference here and what different levels of each means.
77 isnt it better to say photosynthetic ability?!
79 you should state that ROS induce lipid peroxidation. Otherwise there is no connection between the two sentences.
80 please explain why under stress conditions, production of SOD, CAT and POD is increased but their activity (line 83) is reduced.
97 remove “ of the park”. The pond is approximately 300 m long…
99-101 I recommend moving these statements about permits to the acknowledgement sections.
105-106 “… leaves on the water surface have shown an obvious black color in recent years.” I do not understand what the authors are trying to convey here.
108 remove “a location in”.
102-110 in this paragraph, authors could simply say that, in the study site/pond the restoration of V. natans has failed repeatedly despite… list all the failed trials including aerator installation.
111-117 how were these measured? Using what device/probes?
118 black color and anaerobic odor? Is this all we know about the collected sediment/mud?
136-138 The two substrates are being called different things throughout the text. Here is the longest name used have in the text: Bottom mud of the pond and River coarse sand. I suggest calling them simply mud and sand, and at first mention describe how you collected the mud (from the bottom of the pond) and sand. Also provide the chemical and physical properties of both substrates.
141 replace “allayed” with placed.
142 three different depths… what are the depths.
145-148 do you need all these (Temperature, pH, DO, ORP, EC, Turbidity, Chl a (what about Chl b?)) parameters? If you do not discuss them, you should remove them from the text.
153 remove “during the experiment”
153-155 other researchers do not need to know about the permitting agency. Move these to the acknowledgement section.
159-164 rephrase all these sentences. Do not start explaining each method with”By…”. For example, digested water samples were analyzed for TP using molybdenum…
172 “converted to mg g-1 of fresh leaf.” Also how did you measure the fresh weight? Did you ensure that there is no water on the plant tissue when weighting them? How did you deal with the clonal growth? Did you remove them? Are new clonal growth a part of the fresh weight measurement?
193-194 please expand on this statement.
205 statistical analysis
I suggest separating the statistical analysis into two sections and explain the design for lab and field experiments separately. In the result section, you explain the split plot design first but in the stats section, you described it in the latter section. These need to be consistent.
216 was there a depth factor in the aquaria design? I am getting confused.
224 are these reported numbers for both substrates?
227 report nitrate conc at the beginning and at the end.
229 nutrients? Or are you still talking about nitrate? Also, reduction in nutrient may indicate nutrient uptake but not plant growth. Please remove the statement about growth.
232 heterotrophic microbes… please provide supporting literature.
235 change to… was almost double than the growth in sand…
237 is there a statistical analysis done on this part to detect significant differences between the growth of V. natans in the two different substrates?
238-242 start this section by stating that there was a temporal difference between Chl a… content of the leaves and then point at the increased levels by reporting the concentration at the specific days into the experiment.
242 if statistical analysis was done on these measurements, why they are not incorporated in the figures?

246 if both Chl a and b had the same trend, you could revise this paragraph to reflect this observation. That would bes much more effective than making the reader go through them one by one.

248 MDA increased by 5 or 10 d in both leaves and roots. What does this mean?
258 the difference was small? Table 3 shows that there is No difference!
266 how much was the decline? 10%? 50%? Is this a statistically significant decline?
271-274 why are these important? Did temperature and rain both had a significant effect on the experiment? Daily rain and temperature have equal effect (if any) on the treatments. If this is the case, then reporting them does not add to this manuscript and I suggest removing all the statements and fig/tables related to temp and rain.
275 all the water equality parameters changed by day? Did this fluctuation have a sig effect on the treatments? What do you mean by …turbidity were recorded at 0 or 5 d. ? does it mean that they were recorded on 5-day intervals?

335 as mentioned before, the statistical analysis was not shown and it needs to be incorporated in figures or authors provide a table to show there was a statistically significant different between Chl level in leaves in mud than sand.

What was found in the laboratory experiment?

345 what is the hypothesis behind the mud having an adverse effect on the growth of v. natans?
350 I suggest saying that the light quantum or light penetration was reduced by 50% and 90% at depths…
355 sharply increased by 10 d? please mention the fig number. I do not understand what authors are trying to convey here.
359 please cite literature that also reported similar observation under low light conditions instead of chemical exposure. Also what does severe chemical conditions mean?

389 what is the day of peak activity? Please explain?

401-404 this is interesting but should move to the discussion section where you can cite the paper by Han and Cui and discuss their finding. Also this needs to be further discussed since you had a treatment at 0.5 m depth so explain how your treatment performed under this condition.

408 I do not recall where in the text you showed a correlation between rainfall and nutrient concentration! This needs to be removed.

Fig1 mud of pond bottom. Please change to “collected mud”.
Fig2 if these cages were used. Im afraid they might have blocked quite a lot of sunlight themselves!
Fig 3 what is d? please specify how many observation/data point is shown by each circle.
Table 1 the table and its caption does not indicate what each column represents. For example, what are the two “after”? also state the duration of the experiment.
Table 2 how are these measurements done? The numbers seem too clean and all appear to be rounded up. This table is for the lab experiment. What about the field experiment? Did the plants survive 30 day in that depth? How was the growth and other growth components?
Table 3 why was beaker/tray number the random factor and was not considered a replication? You have df for chl a/b in plant organs? This was not measured and there is no statistical analysis on it. Please leave it blank.
Table 4 is confusing. You have different parameters measured on 5-day interval, but at what depth? Then you have different depths that numbers do not differ by much! Should you replace them with a range to simplify the table? The significant difference for depth at the last column, is comparing the three depths (0.5,1.2 and 2 m)?
Why transparency is not measured at different depths? Secchi disk readings could be the most important measurement in this study since you are focusing only on water depth in the field experiment.
What is the + for the TP p value?
All the water quality parameters, those reported in mg/l are not different across depths. why report them for different depths?

Reviewer 2 ·

Basic reporting

The manuscript investigated the effects of substrate type and water depth on the growth and physiological responses of submerged plant Vallisneria natans (Lour.) H.Hara by laboratory and field experiments. The experimental design is quite straightforward. Hence, the study and its results are very easily followed.

Experimental design

In this study, the authors compared the physiological responses of V. natans between bottom mud of the pond and quartz sand. The collected bottom mud was black and emitted anaerobic odors as mentioned by the authors, which may be stressful to V. natans. However, quartz sand may be also bad to the growth of V. natans because of its extremely low nutrition, which means quartz sand may be also stressful to V. natans. Thus, quartz sand may not be suitable as a reference substrate, and using quartz sand as a reference substrate can not achieve the purpose of the study. The authors should notice the limitation and should be more cautious when discussing the results. If I did this experiment, I would choose another suitable substrate as a reference which lets V. natans growth well in the natural habitats (such as a mixture of sand and soil, please see the Xiao et al., 2007 in Hydrobiologia, the authors also cited in the text).

Validity of the findings

The authors investigated the physiological status of V. natans in response to substrate type and water depth in the field experiment, but they only compared the physiological responses of V. natans between two substrate types in laboratory experiment. The missed data about the response of V. natans to water depth may hinder the comprehensive understanding of the physiological status V. natans in response to substrate type and water depth.

Additional comments

The manuscript investigated the effects of substrate type and water depth on the growth and physiological responses of submerged plant Vallisneria natans (Lour.) H.Hara by laboratory and field experiments. The experimental design is quite straightforward. Hence, the study and its results are very easily followed. However, I still have some concerns that the authors need to deal with:
1. In this study, the authors compared the physiological responses of V. natans between bottom mud of the pond and quartz sand. The collected bottom mud was black and emitted anaerobic odors as mentioned by the authors, which may be stressful to V. natans. However, quartz sand may be also bad to the growth of V. natans because of its extremely low nutrition, which means quartz sand may be also stressful to V. natans. Thus, quartz sand may not be suitable as a reference substrate, and using quartz sand as a reference substrate can not achieve the purpose of the study. The authors should notice the limitation and should be more cautious when discussing the results. If I did this experiment, I would choose another suitable substrate as a reference which lets V. natans growth well in the natural habitats (such as a mixture of sand and soil, please see the Xiao et al., 2007 in Hydrobiologia, the authors also cited in the text).
2. The authors investigated the physiological status of V. natans in response to substrate type and water depth in the field experiment, but they only compared the physiological responses of V. natans between two substrate types in laboratory experiment. The missed data about the response of V. natans to water depth may hinder the comprehensive understanding of the physiological status V. natans in response to substrate type and water depth.
3. Specific comments:
Long tile, I suggest delete ‘a useful aquatic plant for water environment restoration in urban areas’
Line124-125. Please clarify how many repetitions in laboratory experiment? A replication for each beaker, or each aquarium?
Line132. 500 L of water was sampled from each aquarium??? Or 500 mL?
Lin136-142. Again, how many repetitions in field experiment? Please clarify it.
Line 208. How could the experimental time be treated as a fixed factor? Experimental time should be a repeated factor, and thus a two-way repeated ANOVA should be conducted to investigate the effects of experiment time and depth on the variables measured by the water quality meter (i.e., temperature, pH, DO, ORP, EC, turbidity, and Chl-a).
Line 234. If the plant biomass included the biomass sampled for physiological measurements? Please clarify.
For the Results, there are too many tables and figures, I strongly suggest the authors do some consolidation and streamlining.
For the Discussion, the authors mainly listed the results but did not explain why these results occurred, and what is the ecological significance of the obtained results. Moreover, the authors discussed the effects of substrate type and water depth separately, but not mentioned their interactive effects which may be most interesting results!

The language is in general easy to understand, but there are minor problems throughout the text (according to my view, but I am not a native English speaker). The authors need to check the text carefully, and ideally also ask a native speaker to have a quick check.

---

## Round 0.2 · Minor Revisions

Reviewer 2 identified one remaining issue which should be resolved before accepting your manuscript. I am looking forward to the revised manuscript.

Reviewer 2 ·

Basic reporting

no comment

Experimental design

no comment

Validity of the findings

no comment

Additional comments

The authors have made substantial changes according to my comments and addressed all my questions.I am generally satisfied with the revised version.The only question is that authors investigated the physiological status of V. natans in response to substrate type and water depth (two factors) in the field experiment, but they only compared the physiological responses of V. natans between two substrate types (one factor) in laboratory experiment. The two experiments can not be well matched and complement each other. The authors should notice the limitation.

---

## Round 0.3 · accepted · Accept

Thank you very much for taking up the reviewers' suggestions constructively and implementing them to improve the manuscript.